# EMPIRICAL FREQUENTIST COVERAGE OF DEEP LEARNING UNCERTAINTY QUANTIFICATION PROCEDURES

## ABSTRACT

Uncertainty quantification for complex deep learning models is increasingly important as these techniques see growing use in high-stakes, real-world settings. Currently, the quality of a model's uncertainty is evaluated using point-prediction metrics such as negative log-likelihood or the Brier score on heldout data. In this study, we provide the first large scale evaluation of the empirical frequentist coverage properties of well known uncertainty quantification techniques on a suite of regression and classification tasks. We find that, in general, some methods do achieve desirable coverage properties on *in distribution* samples, but that coverage is not maintained on out-of-distribution data. Our results demonstrate the failings of current uncertainty quantification techniques as dataset shift increases and establish coverage as an important metric in developing models for real-world applications.

## 1 INTRODUCTION

Predictive models based on deep learning have seen dramatic improvement in recent years (LeCun et al., 2015), which has led to widespread adoption in many areas. For critical, high-stakes domains such as medicine or self-driving cars, it is imperative that mechanisms are in place to ensure safe and reliable operation. Crucial to the notion of safe and reliable deep learning is the effective quantification and communication of *predictive uncertainty* to potential end-users of a system. Many approaches have recently been proposed that fall into two broad categories: ensembles and Bayesian methods. Ensembles (Lakshminarayanan et al., 2017) aggregate information from many individual models to provide a measure of uncertainty that reflects the ensembles agreement about a given data point. Bayesian methods offer direct access to predictive uncertainty through the posterior predictive distribution, which combines prior knowledge with the observed data. Although conceptually elegant, calculating exact posteriors of even simple neural models is computationally intractable (Yao et al., 2019; Neal, 1996), and many approximations have been developed (Hernández-Lobato & Adams, 2015; Blundell et al., 2015; Graves, 2011; Pawlowski et al., 2017; Hernández-Lobato et al., 2016; Louizos & Welling, 2016; 2017). Though approximate Bayesian methods scale to modern sized data and models, recent work has questioned the quality of the uncertainty provided by these approximations (Yao et al., 2019; Wenzel et al., 2020; Ovadia et al., 2019).

Previous work assessing the quality of uncertainty estimates have focused on calibration metrics and scoring rules such as the negative-loglikelihood (NLL), expected calibration error (ECE), and Brier score. Here we provide a complementary perspective based on the notion of empirical *coverage*, a well-established concept in the statistical literature (Wasserman, 2013) that evaluates the quality of a predictive *set* or *interval* instead of a point prediction. Informally, coverage asks the question: If a model produces a predictive uncertainty interval, how often does that interval actually contain the observed value? Ideally, predictions on examples for which a model is uncertain would produce larger intervals and thus be more likely to cover the observed value. More formally, given features $\mathbf{x_n} \in \mathbb{R}^d$ and a response $y_n \in \mathbb{R}$, coverage is defined in terms of a set $\hat{\mathcal{C}}_n(\mathbf{x})$ and a level $\alpha \in [0, 1]$. The set $\hat{\mathcal{C}}_n(\mathbf{x})$ is said to have coverage at the $1 - \alpha$ level if for all distributions $P \in \mathbb{R}^d \times \mathbb{R}$ where $(\mathbf{x}, y) \sim P$, the following inequality holds:

$$\mathbb{P}\{y_n \in \hat{\mathcal{C}}_n(\mathbf{x_n})\} \geq 1 - \alpha \qquad (1)$$

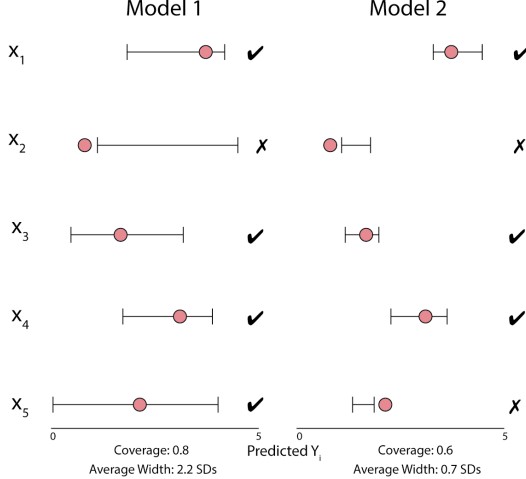

Figure 1: An example of the coverage properties for two methods of uncertainty quantification. In this scenario, each model produces an uncertainty interval for each $x_i$ which attempts to cover the true $y_i$, represented by the red points. Coverage is calculated as the fraction of true values contained in these regions, while the width of these regions is reported in terms of multiples of the standard deviation of the training set $y_i$ values.

The set $\hat{\mathcal{C}}_n(\mathbf{x})$ can be constructed using a variety of procedures. For example, in the case of simple linear regression a prediction interval for a new point $x_{n+1}$ can be constructed[1] using a simple, closed-form solution. Figure 1 provides a graphical depiction of coverage for two hypothetical regression models.

A complementary metric to coverage is *width*, which is the size of the prediction interval or set. Width can provide a relative ranking of different methods, i.e. given two methods with the same level of coverage we should prefer the method that provides intervals with smaller widths.

**Contributions:** In this study we investigate the empirical coverage properties of prediction intervals constructed from a catalog of popular uncertainty quantification techniques such as ensembling, Monte-Carlo dropout, Gaussian processes, and stochastic variational inference. We assess the coverage properties of these methods on nine regression tasks and two classification tasks with and without dataset shift. These tasks help us make the following contributions:

- We introduce coverage and width as a natural and interpretable metrics for evaluating predictive uncertainty.

- A comprehensive set of coverage evaluations on a suite of popular uncertainty quantification techniques.

- An examination of how dataset shift affects these coverage properties.

## 2 BACKGROUND AND RELATED WORK

**Obtaining Predictive Uncertainty Estimates**
Several lines of work focus on improving approximations of the posterior of a Bayesian neural network (Graves, 2011; Hernández-Lobato & Adams, 2015; Blundell et al., 2015; Hernández-Lobato et al., 2016; Louizos & Welling, 2016; Pawlowski et al., 2017; Louizos & Welling, 2017). Yao et al.

---

[1] A well-known result from the statistics literature (c.f. chapter 13 of Wasserman (2013)) is that the interval is given by $\hat{y}_{n+1} \pm t_{n-2} s_y \sqrt{1/n + (x_{n+1} - \bar{x})^2/((n-1)s_x^2)}$, where $\hat{y}_{n+1}$ is the predicted value, $t_{n-2}$ is the $1 - \alpha/2$ critical value from a t-distribution with $n - 2$ degrees of freedom, $\bar{x}$ is the mean of $x$ in the training data, and $s_y, s_x$ are the standard deviations for $y$ and $x$ respectively. such that (1) holds asymptotically. However, for more complicated models such as deep learning, closed form solutions with coverage guarantees are unavailable, and constructing these intervals via the bootstrap (Efron, 1982)) can be computationally infeasible or fail to provide the correct coverage (Chatterjee & Lahiri, 2011).

(2019) provide a comparison of many of these methods and highlight issues with common metrics of comparison, such as test-set log likelihood and RMSE. Good scores on these metrics often indicates that the model posterior happens to match the test data rather than the true posterior (Yao et al., 2019). Maddox et al. (2019) developed a technique to sample the approximate posterior from the first moment of SGD iterates. Wenzel et al. (2020) demonstrated that despite advances in these approximations, there are still outstanding challenges with Bayesian modeling for deep networks.

Alternative methods that do not rely on estimating a posterior over the weights of a model can also be used to provide uncertainty estimates. Gal & Ghahramani (2016), for instance, demonstrated that Monte Carlo dropout is related to a variational approximation to the Bayesian posterior implied by the dropout procedure. Lakshminarayanan et al. (2017) used ensembling of several neural networks to obtain uncertainty estimates. Guo et al. (2017) established that temperature scaling provides well calibrated predictions on an i.i.d test set. More recently, van Amersfoort et al. (2020) showed that the distance from the centroids in a RBF neural network yields high quality uncertainty estimates. Liu et al. (2020) also leveraged the notion of distance (in this case, the distance from test to train examples) to obtain uncertainty estimates with their Spectral-normalized Neural Gaussian Processes.

**Assessments of Uncertainty Properties under Dataset Shift**
Ovadia et al. (2019) analyzed the effect of dataset shift on the accuracy and calibration of Bayesian deep learning methods. Their large scale empirical study assessed these methods on standard datasets such as MNIST, CIFAR-10, ImageNet, and other non-image based datasets. Additionally, they used translations, rotations, and corruptions (Hendrycks & Gimpel, 2017) of these datasets to quantify performance under dataset shift. They found stochastic variational inference (SVI) to be promising on simpler datasets such as MNIST and CIFAR-10, but more difficult to train on larger datasets. Deep ensembles had the most robust response to dataset shift.

**Theoretical Coverage Guarantees**
The Bernstein-von Mises theorem connects Bayesian credible sets and frequentist confidence intervals. Under certain conditions, Bayesian credible sets of level $\alpha$ are asymptotically frequentist confidence sets of level $\alpha$ and thus have the same coverage properties. However, when there is model misspecification, coverage properties no longer hold (Kleijn & van der Vaart, 2012).

Barber et al. (2019) explored under what conditions conditional coverage guarantees can hold for arbitrary models (i.e. guarantees for $\mathbb{P}\{y_n \in \hat{C}_n(\mathbf{x}|\mathbf{x} = \mathbf{x_n})\}$, which are *per sample* guarantees). They show that even when these coverage properties are not desired to hold for any possible distribution, there are provably no methods that can give such guarantees. By extension, no Bayesian deep learning methods can provide conditional coverage guarantees.

## 3 METHODS

In both the regression and classification settings, we analyzed the coverage properties of prediction intervals and sets of five different approximate Bayesian and non-Bayesian approaches for uncertainty quantification. These include Dropout (Gal & Ghahramani, 2016; Srivastava et al., 2015), ensembles (Lakshminarayanan et al., 2017), Stochastic Variational Inference (Blundell et al., 2015; Graves, 2011; Louizos & Welling, 2016; 2017; Wen et al., 2018), and last layer approximations of SVI and Dropout (Riquelme et al., 2019). Additionally, we considered prediction intervals from linear regression and the 95% credible interval of a Gaussian process with the squared exponential kernel as baselines in regression tasks. For classification, we also considered temperature scaling (Guo et al., 2017) and the softmax output of vanilla deep networks (Hendrycks & Gimpel, 2017).

### 3.1 REGRESSION METHODS AND METRICS

We evaluated the coverage properties of these methods on nine large real world regression datasets used as a benchmark in Hernández-Lobato & Adams (2015) and later Gal and Ghahramani (Gal & Ghahramani, 2016). We used the training, validation, and testing splits publicly available from Gal and Ghahramani and performed nested cross validation to find hyperparameters and evaluated coverage properties, defined as the fraction of prediction intervals which contained the true value in the test set. On the training sets, we did 100 trials of a random search over hyperparameter space of a multi-layer-perceptron architecture with an Adam optimizer (Kingma & Ba, 2015) and selected hyperparameters based on RMSE on the validation set.

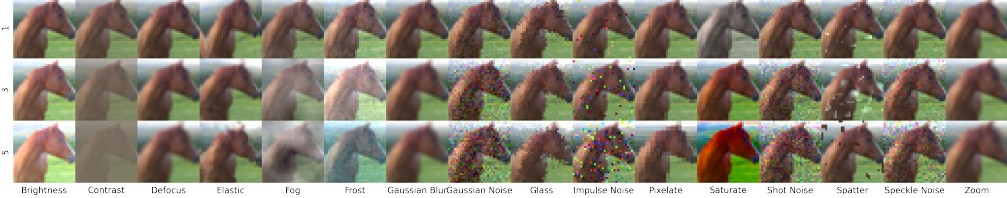

Figure 2: An example of the corruptions in CIFAR-10-C. The 16 different corruptions have 5 discrete levels of shift, of which 3 are shown here. The same corruptions were applied to ImageNet to form the ImageNet-C dataset.

Each approach required slightly different ways to obtain a 95% prediction interval. For an ensemble of neural networks, we trained $N = 40$ vanilla networks and used the 2.5% and 97.5% quantiles as the boundaries of the prediction interval. For dropout and last layer dropout, we made 200 predictions per sample and similarly discarded the top and bottom 2.5% quantiles. For SVI, last layer SVI (LL SVI), and Gaussian processes we had approximate variances available for the posterior which we used to calculate the prediction interval. We calculated 95% prediction intervals from linear regression using the closed-form solution.

Then we calculated two metrics:
- **Coverage**: A sample is considered covered if the true label is contained in this 95% prediction interval. We average over all samples in a test set to estimate the coverage of a method on this dataset.
- **Width**: The width is the average over the test set of the ranges of the 95% prediction intervals.

Coverage measures how often the true label is in the prediction region while width measures how specific that prediction region is. Ideally, we would have high levels of coverage with low levels of width on in-distribution data. As data becomes increasingly out of distribution, we would like coverage to remain high while width increases to indicate model uncertainty.

## 3.2 CLASSIFICATION METHODS AND METRICS

Ovadia et al. (2019) evaluated model uncertainty on a variety of datasets publicly available. These predictions were made with the five apprxoimate Bayesian methods describe above, plus vanilla neural networks, with and without temperature scaling. We focus on the predictions from MNIST, CIFAR-10, CIFAR-10-C, ImageNet, and ImageNet-C datasets. For MNIST, Ovadia et al. (2019) measured model predictions on rotated and translated versions of the test set. For CIFAR-10, Ovadia et al. (2019) measured model predictions on translated and corrupted versions of the test set. For ImageNet, Ovadia et al. (2019) only analyzed model predictions on the corrupted images of ImageNet-C. Each of these transformations (rotation, translation, or any of the 16 corruptions) has multiple levels of shift. Rotations range from 15 to 180 degrees in 15 degrees increments. Translations shift images every 2 and 4 pixels for MNIST and CIFAR-10, respectively. Corruptions have 5 increasing levels of intensity. Figure 2 shows the effects of the 16 corruptions in CIFAR-10-C at the first, third, and fifth levels of intensity.

We calculate the prediction set of a model's output. Given $\alpha \in (0, 1)$, the $1 - \alpha$ prediction set $\mathcal{S}$ for a sample $\mathbf{x_i}$ is the minimum sized set of classes such that

$$\sum_{c \in \mathcal{S}} p(y_c | \mathbf{x_i}) \geq 1 - \alpha \tag{2}$$

This consists of the top $k_i$ probabilities such that $1 - \alpha$ probability has been accumulated. Then we can define:
- **Coverage:** For each dataset point, we calculate the $1 - \alpha$ prediction set of the label probabilities, then coverage is what fraction of these prediction sets contain the true label.
- **Width:** The width of a prediction set is simply the number of labels in the set, $|\mathcal{S}|$. We report the average width of prediction sets over a dataset in our figures.

Although both calibration (Guo et al., 2017) and coverage can involve a probability over a model's output, calibration only considers the most likely label and it's corresponding probability, while coverage considers the the top-$k_i$ probabilities. In the classification setting, coverage is more robust to label errors as it does not penalize models for putting probability on similar classes.

# 4 RESULTS

## 4.1 REGRESSION

Table 1 shows the mean and standard error of coverage levels for the methods we evaluated. In the regression setting, we find high levels of coverage for linear regression, Gaussian processes, SVI, and LL SVI. Ensembles and Dropout had lower levels of coverage, while LL Dropout had the lowest average coverage. Table 2 reports the average width of the 95% prediction interval in terms of standard deviations of the response variable. We see that higher coverage correlates with a higher average width.

| Dataset \| Method | Linear Regression | GP | Ensemble | Dropout | LL Dropout | SVI | LL SVI |
|---|---|---|---|---|---|---|---|
| Boston Housing | 0.9461 (5.61e-03) | **0.9765 (5.05e-03)** | 0.5912 (1.43e-02) | 0.602 (1.64e-02) | 0.1902 (2.01e-02) | 0.9434 (6.04e-03) | 0.9339 (8.48e-03) |
| Concrete | 0.9437 (2.68e-03) | **0.967 (3.02e-03)** | 0.5854 (1.04e-02) | 0.7282 (1.17e-02) | 0.0932 (1.75e-02) | 0.9581 (3.61e-03) | 0.9443 (6.72e-03) |
| Energy | 0.8957 (4.66e-03) | 0.8857 (6.96e-03) | 0.8669 (5.26e-03) | 0.8013 (2.00e-02) | 0.2597 (2.75e-02) | 0.9773 (3.02e-03) | **0.9938 (2.99e-03)** |
| Kin8nm | 0.9514 (1.20e-03) | **0.9705 (1.53e-04)** | 0.6706 (4.43e-03) | 0.8037 (8.15e-03) | 0.1984 (1.36e-02) | 0.9618 (2.63e-03) | 0.9633 (1.36e-03) |
| Naval Propulsion Plant | 0.9373 (1.59e-03) | **0.9994 (2.12e-04)** | 0.8036 (5.99e-03) | 0.9212 (6.76e-03) | 0.2683 (2.51e-02) | 0.9797 (1.88e-03) | 0.9941 (1.25e-03) |
| Power Plant | **0.9646 (1.14e-03)** | 0.9614 (1.26e-03) | 0.4008 (1.12e-02) | 0.432 (1.47e-02) | 0.1138 (1.41e-02) | 0.9626 (1.13e-03) | 0.9623 (1.60e-03) |
| Protein Tertiary Structure | **0.9619 (4.71e-04)** | 0.959 (4.72E-04) | 0.4125 (2.98e-03) | 0.3846 (1.36e-02) | 0.1182 (1.35e-02) | 0.9609 (2.27e-03) | 0.9559 (1.72e-03) |
| Wine Quality Red | 0.9425 (2.32e-03) | **0.9472 (3.28e-03)** | 0.3919 (1.18e-02) | 0.3566 (1.83e-02) | 0.1616 (7.45e-03) | 0.9059 (8.19e-03) | 0.8647 (8.77e-03) |
| Yacht Hydrodynamics | 0.9449 (7.86e-03) | 0.9726 (6.73e-03) | 0.9161 (7.38e-03) | 0.3871 (2.82e-02) | 0.2081 (2.54e-02) | 0.9807 (6.97e-03) | **0.9899 (6.03e-03)** |

Table 1: The average coverage of six methods across nine datasets with the standard error over 20 cross validation folds in parentheses.

| Dataset \| Method | Linear Regression | GP | Ensemble | Dropout | LL Dropout | SVI | LL SVI |
|---|---|---|---|---|---|---|---|
| Boston Housing | 2.0424 (6.87E-03) | 1.8716 (1.17E-02) | 0.4432 (7.82E-03) | 0.6882 (2.19E-02) | 0.1855 (2.05E-02) | 1.301 (2.56E-02) | 1.148 (2.36E-02) |
| Concrete | 2.4562 (2.22E-03) | 2 (3.32E-03) | 0.4776 (9.03E-03) | 1.0342 (1.79E-02) | 0.1028 (2.04E-02) | 1.5116 (1.72E-02) | 1.2293 (1.41E-02) |
| Energy | 1.144 (2.29E-03) | 1.0773 (2.64E-03) | 0.2394 (2.56E-03) | 0.5928 (1.22E-02) | 0.1417 (1.61E-02) | 0.8426 (1.73E-02) | 0.7974 (1.95E-02) |
| Kin8nm | 3.0039 (9.76E-04) | 2.3795 (7.02E-03) | 0.5493 (2.37E-03) | 1.2355 (1.37E-02) | 0.2024 (1.22E-02) | 1.6697 (7.75E-03) | 1.2624 (2.99E-03) |
| Naval Propulsion Plant | 1.5551 (7.12E-04) | 0.3403 (1.00E-02) | 0.6048 (4.86E-03) | 1.1593 (6.45E-03) | 0.2281 (1.83E-02) | 1.3064 (1.38E-01) | 0.488 (5.44E-03) |
| Power Plant | 1.0475 (7.09E-04) | 0.9768 (9.63E-04) | 0.2494 (6.72E-03) | 0.3385 (1.69E-02) | 0.0918 (9.06E-03) | 1.0035 (1.88E-03) | 0.9818 (3.64E-03) |
| Protein Tertiary Structure | 3.3182 (3.21E-04) | 3.2123 (3.47E-03) | 0.6804 (3.77E-03) | 0.9144 (1.41E-02) | 0.3454 (1.99E-02) | 2.9535 (3.82E-02) | 2.6506 (2.20E-02) |
| Wine Quality Red | 3.1573 (1.82E-03) | 3.1629 (4.07E-03) | 0.7763 (1.31E-02) | 0.7841 (2.91E-02) | 0.3481 (1.61E-02) | 2.7469 (2.72E-02) | 2.3597 (2.70E-02) |
| Yacht Hydrodynamics | 2.3636 (2.89E-03) | 1.6974 (7.57E-03) | 0.4475 (9.76E-03) | 0.5443 (2.22E-02) | 0.1081 (9.83E-03) | 0.657 (3.54E-02) | 0.69 (3.81E-02) |

Table 2: The average width of the posterior prediction interval of six methods across nine datasets with the standard error over 20 cross validation folds in parentheses. Width is reported in terms of standard deviations of the response variable in the training set.

| Dataset \| Method | Linear Regression | GP | Ensemble | Dropout | LL Dropout | SVI | LL SVI |
|---|---|---|---|---|---|---|---|
| Boston Housing | 4.0582 (1.22E-01) | 3.5397 (2.30E-01) | 3.1484 (1.31E-01) | 4.9654 (1.27E-01) | 3.6281 (1.61E-01) | **3.148 (1.97E-01)** | 3.4223 (1.93E-01) |
| Concrete | 7.6025 (1.12E-01) | 7.8245 (1.12E-01) | 5.8107 (6.18E-02) | 10.3653 (9.60E-02) | 6.5621 (1.20E-01) | **5.6109 (1.47E-01)** | 6.4618 (1.26E-01) |
| Energy | 2.3029 (6.63E-02) | 2.7454 (5.68E-02) | **1.0912 (1.51E-02)** | 3.172 (6.01E-02) | 1.5211 (6.17E-02) | 1.2781 (6.84E-02) | 2.7032 (6.47E-02) |
| Kin8nm | 0.1199 (8.05E-04) | 0.1366 (1.08E-03) | 0.0855 (2.72E-04) | 0.2027 (5.69E-04) | 0.0984 (1.51E-03) | **0.0816 (5.40E-04)** | 0.1091 (1.47E-03) |
| Naval Propulsion Plant | 0.0054 (4.94E-05) | **6E-04 (3.43E-05)** | 0.0041 (3.08E-05) | 0.0059 (2.29E-05) | 0.006 (6.03E-04) | 0.0012 (4.25E-05) | 0.0042 (5.44E-04) |
| Power Plant | 4.5639 (5.67E-02) | 4.2551 (3.42E-02) | 4.2952 (2.62E-02) | 4.5793 (2.68E-02) | 4.7594 (6.31E-02) | **4.1983 (4.68E-02)** | 4.2903 (3.56E-02) |
| Protein Tertiary Structure | 4.514 (1.06E-02) | 4.9695 (7.61E-03) | **4.2398 (7.03E-03)** | 5.2182 (6.82E-03) | 4.5219 (2.78E-02) | 4.3824 (1.93E-02) | 4.6458 (2.94E-02) |
| Wine Quality Red | 0.6654 (6.56E-03) | 0.6432 (7.79E-03) | **0.643 (6.41E-03)** | 0.664 (5.40E-03) | 0.6555 (6.69E-03) | 0.647 (8.95E-03) | 0.6762 (1.43E-02) |
| Yacht Hydrodynamics | 4.4647 (1.40E-01) | 4.7585 (1.92E-01) | 3.1594 (8.95E-02) | 9.4761 (2.64E-01) | **2.5562 (2.47E-01)** | 2.765 (1.66E-01) | 2.6775 (9.06E-02) |

Table 3: The average RMSE of six methods across nine datasets with the standard error over 20 cross validation folds in parentheses. These values are comparable to other reported in the literature for these benchmarks (Gal & Ghahramani, 2016), though the intention was not to produce state of the art results, but merely demonstrate the models were trained in a reasonable manner.

## 4.2 MNIST

We begin by calculating coverage and width for predictions from Ovadia et al. (2019) on MNIST and shifted MNIST data. Ovadia et al. (2019) used a LeNet architecture and we refer to their manuscript for more details on their implementation.

Figure 3 shows how coverage and width co-vary as dataset shift increases. We observe high coverage and low width for all models on training, validation, and non-shifted test set data. The elevated width for SVI on these dataset splits indicate that the posterior predictions of label probabilities were the most diffuse to begin with among all models. In Figure 3, all seven models have at least 0.95 coverage with a 15 degree rotation shift. Most models don't see an appreciable increase in the average width of the 0.95 prediction set, except for SVI. The average width for SVI jumps to over 2 at 15 degrees rotation. As the amount of shift increases, coverage decreases across all methods in a comparable way. SVI maintains higher levels of coverage, but with a compensatory increase in width.

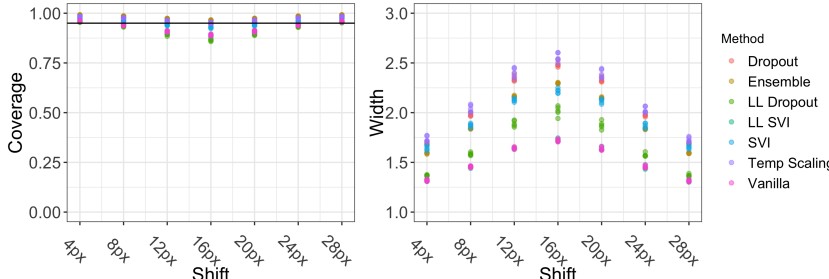

Figure 4: The effect of translation shifts on coverage and width in CIFAR-10 images. Coverage remains robust across all pixel shifts, while width increases.

In Figure 3, we observe the same coverage-width pattern at the lowest level of shift, 2 pixels. All methods have at least 0.95 coverage, but only SVI has a distinct jump in the average width of its prediction set. The average width of the prediction set increases slightly then plateaus for all methods but SVI as the amount of translation increases.

For this simple dataset, SVI outperforms other models with regards to coverage and width properties. It is the only model that has an average width that corresponds to the amount of shift observed.

| Method | Mean Test Set Coverage (SE) | Mean Test Set Width (SE) | Mean Rotation Shift Coverage (SE) | Mean Rotation Shift Width (SE) | Mean Translation Shift Coverage (SE) | Mean Translation Shift Width (SE) |
|---|---|---|---|---|---|---|
| Dropout | 0.9987 (6.32E-05) | 1.06 (1.38E-04) | 0.5519 (2.91E-02) | 2.3279 (6.64E-02) | 0.5333 (3.54E-02) | 2.3527 (6.34E-02) |
| Ensemble | 0.9984 (7.07E-05) | 1.0424 (2.07E-04) | 0.5157 (3.11E-02) | 2.0892 (5.44E-02) | 0.5424 (3.33E-02) | 2.3276 (6.66E-02) |
| LL Dropout | 0.9985 (1.05E-04) | 1.0561 (1.89E-03) | 0.552 (2.93E-02) | 2.3162 (6.73E-02) | 0.5388 (3.52E-02) | 2.3658 (6.66E-02) |
| LL SVI | 0.9984 (1.14E-04) | 1.0637 (1.65E-03) | 0.5746 (2.77E-02) | 2.6324 (8.41E-02) | 0.535 (3.51E-02) | 2.3294 (6.46E-02) |
| SVI | 0.9997 (7.35E-05) | 1.5492 (2.19E-02) | 0.7148 (2.06E-02) | 4.8549 (1.44E-01) | 0.754 (1.96E-02) | 5.6803 (1.99E-01) |
| Temp scaling | 0.9986 (1.36E-04) | 1.0642 (1.98E-03) | 0.5243 (3.10E-02) | 2.2683 (6.17E-02) | 0.5375 (3.33E-02) | 2.347 (6.21E-02) |
| Vanilla | 0.9972 (1.16E-04) | 1.032 (9.06E-04) | 0.4715 (3.28E-02) | 1.7492 (3.78E-02) | 0.4798 (3.50E-02) | 1.801 (3.84E-02) |

Table 4: MNIST average coverage and width for the test set, rotation shift, and translation shift.

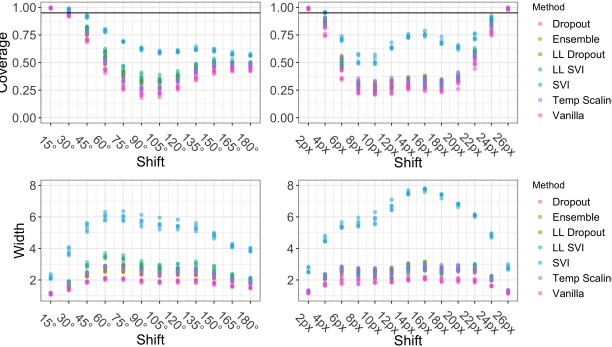

Figure 3: The effect of rotation and translation on coverage and width, respectively, for MNIST.

## 4.3 CIFAR-10

Next, we consider a more complex image dataset, CIFAR-10. Ovadia et al. (2019) trained 20 layer and 50 layer ResNets. Figure 4 shows how all seven models have high coverage levels over all translation shifts. Temperature scaling and ensemble, in particular, have at least 0.95 coverage for every translation. We find that this high coverage comes with increases in width as shift increases. Figure 4 shows that temperature scaling has the highest average width across all models and shifts. All models have the same pattern of width increases, with peak average widths at 16 pixels translation.

Between the models which satisfy 0.95 coverage levels on all shifts, ensemble models have lower width than temperature scaling models. Under translation shifts on CIFAR-10, ensemble methods perform the best given their high coverage and lower width.

Additionally, we consider the coverage properties of models on 16 different corruptions of CIFAR-10 from Hendrycks and Gimpel (Hendrycks & Gimpel, 2017). Figure 5 shows coverage vs. width over varying levels of shift intensity. Models that have more dispersed points to the right have higher

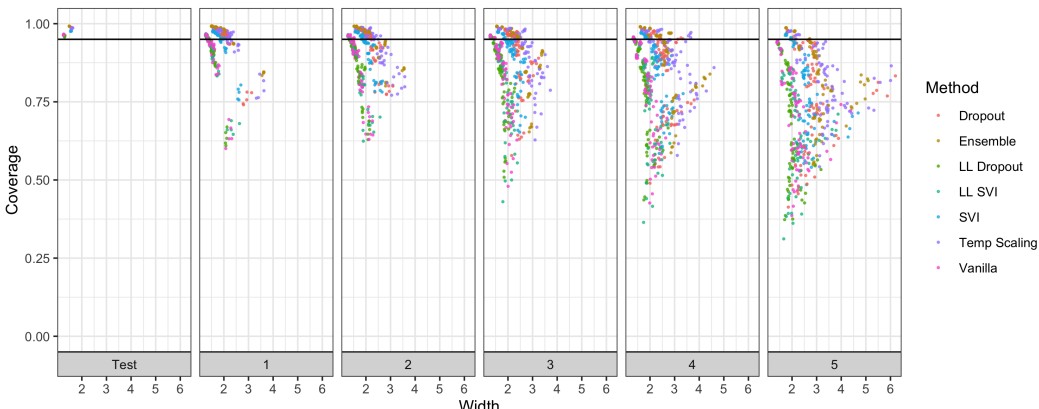

Figure 5: The effect of corruption intensity on coverage levels vs. width in CIFAR-10-C. Each facet panel represents a different corruption level, while points are the coverage of a model on one of 16 corruptions. Each facet has 80 points per method, since 5 iterations were trained per method. For methods with points at the same coverage level, the superior method is to the left as it has a lower width.

widths for the same level of coverage. An ideal model would have a cluster of points above the 0.95 coverage line and be far to the left portion of each facet. For models that have similar levels of coverage, the superior method will have points further to the left.

Figure 5 demonstrates that at the lowest shift intensity, ensemble models, dropout, temperature scaling, and SVI were able to generally provide high levels of coverage on most corruption types. However, as the intensity of the shift increases, coverage decreases. Ensembles and dropout models have for at least half of their 80 model-corruption evaluations at least 0.95 coverage up to the third intensity level. At higher levels of shift intensity, ensembles, dropout, and temperature scaling consistently have the highest levels of coverage. Although these higher performing methods have similar levels of coverage, they have different widths. See Figure A1 for a further examination of the coverage and widths of these methods.

| Method | Mean Test Set Coverage (SE) | Mean Test Set Width (SE) | Mean Corruption Coverage (SE) | Mean Corruption Width (SE) |
|---|---|---|---|---|
| Dropout | 0.987 (3.72E-04) | 1.578 (2.68E-03) | 0.886 (6.34E-03) | 2.313 (3.03E-02) |
| Ensemble | 0.992 (9.70E-05) | 1.492 (1.52E-03) | 0.911 (5.16E-03) | 2.425 (3.69E-02) |
| LL Dropout | 0.960 (8.77E-04) | 1.301 (3.99E-03) | 0.815 (7.48E-03) | 1.699 (1.53E-02) |
| LL SVI | 0.964 (6.64E-04) | 1.258 (2.60E-03) | 0.817 (7.52E-03) | 1.781 (2.15E-02) |
| SVI | 0.976 (5.10E-04) | 1.558 (6.31E-03) | 0.881 (5.45E-03) | 2.161 (2.32E-02) |
| Temp Scaling | 0.985 (4.54E-04) | 1.599 (1.19E-02) | 0.899 (4.85E-03) | 2.636 (3.86E-02) |
| Vanilla | 0.964 (6.36E-04) | 1.261(3.90E-03) | 0.823 (7.10E-03) | 1.790 (2.16E-02) |

Table 5: The mean coverage and widths on the test set of CIFAR-10 as well as on the mean coverage and width averaged over 16 corruptions and 5 intensities.

## 4.4 IMAGENET

Finally, we analyze coverage and width on ImageNet and ImageNet-C from Hendrycks & Gimpel (2017). Figure 6 shows similar coverage vs. width plots to Figure 5. We find that over the 16 different corruptions at 5 levels, ensembles, temperature scaling, and dropout models had consistently higher levels of coverage. Unsurprisingly, Figure 6 shows that these methods have correspondingly higher widths. At the first three levels of corruption, ensembling has the lowest level of width of the top performing methods (see Figure A2). However, at the highest two levels of corruption, dropout has lower width than ensembling. None of the methods have a commensurate increase in width to maintain the 0.95 coverage levels seen on in-distribution test data as dataset shift increases.

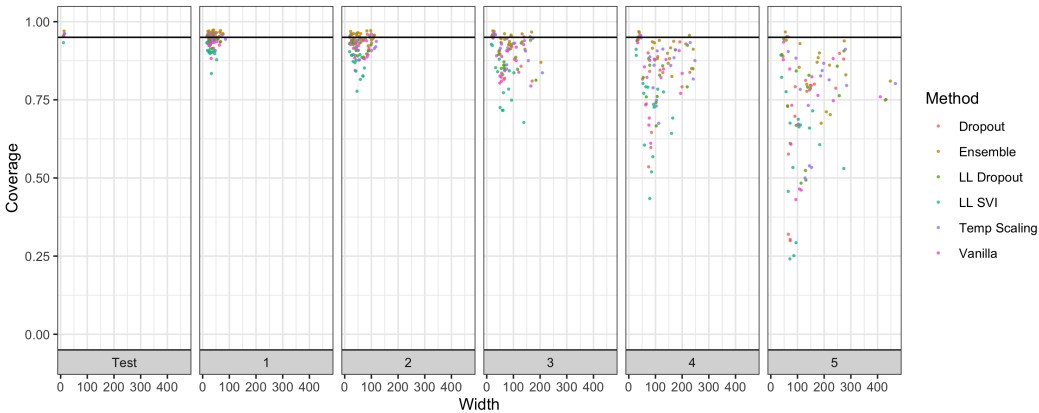

Figure 6: The effect of corruption intensity on coverage levels vs. width in ImageNet-C. Each facet panel represents a different corruption level, while points are the coverage of a model on one of 16 corruptions. Each facet has 16 points per method, as only only 1 iteration was trained per method. For methods equal coverage, the superior method is to the left as it has a lower width.

| Method | Mean Test Set Coverage | Mean Test Set Width | Mean Corruption Coverage (SE) | Mean Corruption Width (SE) |
|---|---|---|---|---|
| Dropout | 0.9613 | 13.2699 | 0.8579 (1.61E-02) | 87.5784 (7.80E+00) |
| Ensemble | 0.9701 | 13.0613 | 0.9231 (7.13E-03) | 105.3608 (8.57E+00) |
| LL Dropout | 0.9552 | 10.7707 | 0.8688 (1.18E-02) | 88.0326 (8.04E+00) |
| LL SVI | 0.9327 | 10.5624 | 0.777 (1.76E-02) | 65.9982 (5.01E+00) |
| Temp Scaling | 0.9613 | 15.4811 | 0.8829 (1.10E-02) | 105.1409 (8.43E+00) |
| Vanilla | 0.9525 | 11.0255 | 0.8529 (1.27E-02) | 80.687 (7.16E+00) |

Table 6: The mean coverage and widths on the test set of ImageNet as well as on the mean coverage and width averaged over 16 corruptions and 5 intensities.

## 5 DISCUSSION

We have provided the first comprehensive empirical study of the frequentist-style coverage properties of popular uncertainty quantification techniques for deep learning models. In regression tasks, Gaussian Processes were the clear winner in terms of coverage across nearly all benchmarks, with smaller widths than linear regression, whose prediction intervals come with formal guarantees. SVI and LL SVI also had excellent coverage properties across most tasks with tighter intervals than GPs and linear regression. In contrast, the methods based on ensembles and Monte Carlo dropout had significantly worse coverage due to their overly confident and tight prediction intervals. Another interesting finding is that despite higher levels of uncertainty (e.g. larger widths), SVI was also the most accurate model based on RMSE as reported in Table 3.

In the classification setting, all methods showed very high coverage in the i.i.d setting (i.e. no dataset shift), as coverage is reflective of top-1 accuracy in this scenario. On MNIST data, SVI had the best performance, maintaining high levels of coverage under slight dataset shift and scaling the width of its prediction intervals more appropriately as shift increased relative to other methods. On CIFAR-10 data, ensemble models were superior. They had the highest levels of coverage at the third of five intensity levels on CIFAR-10-C data, while have lower width than the next best method, temperature scaling. Dropout and SVI also had slightly worse coverage levels, but lower widths as well. Last layer dropout and last layer SVI performed poorly, oftentimes having lower coverage than vanilla neural networks.

In summary, we find that popular uncertainty quantification methods for deep learning models do not provide good coverage properties under moderate levels of datset shift. Although the width of prediction regions do increase under increasing amounts of shift, these changes are not enough to maintain the levels of coverage seen on i.i.d data. We conclude that the methods we evaluated for uncertainty quantification are likely insufficient for use in high-stakes, real-world applications where dataset shift is likely to occur.

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

## A  APPENDIX

**Code Availability**
The code and data to reproduce our results will be made available after the anonymous review period.

| Method | Mean Test Set Coverage (SE) | Mean Test Set Width (SE) | Mean Translation Shift Coverage (SE) | Mean Translation Shift Width (SE) |
|---|---|---|---|---|
| Dropout | 0.9883 (3.79E-04) | 1.5778 (2.68E-03) | 0.9696 (2.48E-03) | 2.0709 (5.11E-02) |
| Ensemble | 0.9922 (3.08E-04) | 1.4925 (1.52E-03) | 0.9806 (1.65E-03) | 1.9246 (4.49E-02) |
| LL Dropout | 0.9628 (1.40E-03) | 1.3007 (3.99E-03) | 0.9184 (5.59E-03) | 1.6678 (4.16E-02) |
| LL SVI | 0.9677 (1.10E-03) | 1.2585 (2.60E-03) | 0.929 (4.55E-03) | 1.5044 (2.61E-02) |
| SVI | 0.9789 (6.41E-04) | 1.5579 (6.31E-03) | 0.9543 (2.89E-03) | 1.9286 (3.69E-02) |
| Temp scaling | 0.9871 (3.51E-04) | 1.5987 (1.19E-02) | 0.9707 (1.97E-03) | 2.1266 (5.30E-02) |
| Vanilla | 0.9686 (6.06E-04) | 1.2611 (3.90E-03) | 0.9296 (4.36E-03) | 1.5064 (2.58E-02) |

Table A1: CIFAR-10 average coverage and width for the test set and translation shift.

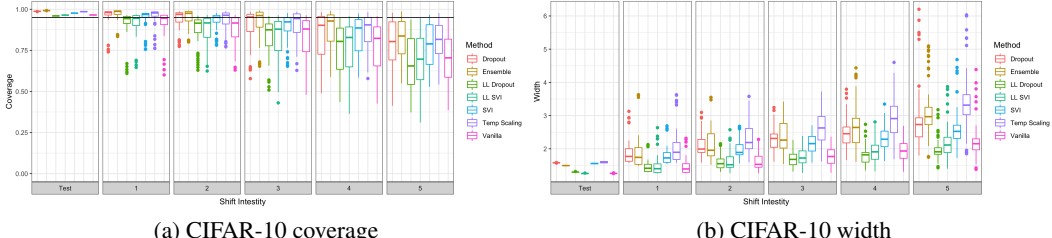

(a) CIFAR-10 coverage                          (b) CIFAR-10 width

Figure A1: The effect of corruption intensity on coverage levels in CIFAR-10. This is averaged over 16 different corruption types. As shift intensity increases, coverage decreases and width increases. In general, Dropout, ensembling, and temperature scaling have the highest levels of coverage across corruptions levels.

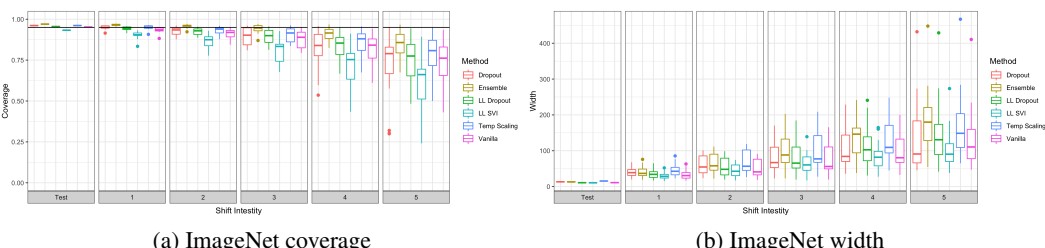

(a) ImageNet coverage                          (b) ImageNet width

Figure A2: The effect of corruption intensity on coverage levels in ImageNet. This is averaged over 16 different corruption types. As shift intensity increases, coverage decreases and width increases. In general, Dropout, ensembling, and temperature scaling have the highest levels of coverage across corruptions levels.

