# OpenReview forum: "Empirical Frequentist Coverage of Deep Learning Uncertainty Quantification Procedures"
_ICLR.cc/2021/Conference — Reject_

### Official Review · AnonReviewer1 · 2020-10-15
**Confused definition of coverage**

**Rating:** 3
**Confidence:** 5

**Review:**

In the submitted manuscript, "Empirical Frequentist coverage of deep learning uncertainty quantification procedures", the authors propose to investigate the Frequentist coverage properties of predictive intervals by numerical experiment for a number of machine learning models applied to benchmark datasets.  I can't say that I find this a strong submission because:
1. the authors give a confused (mis-)definition of coverage; essentially they seem to have taken Barber et al.'s definition of "marginal distribution free prediction intervals", mangled it and then called it Frequentist coverage citing Wasserman
2. the authors claim one of the contributions of this manuscript to be "introduce coverage and width as a natural and interpretable metrics for evaluation predictive uncertainty" but in fact these aspects of predictive intervals from ML models has been studied for many years, as a simple google search will confirm
3. the results shown will not generalise in any meaningful sense: for example, GPs are found to have excellent coverage over the set of regression tasks shown, but in fact GPs are themselves a case study in the difficulties of achieving Frequentist style coverage in the domain of Bayesian non-parametrics (e.g. Hadji & Szabo 2019; Neiswanger & Ramdas 2020; Rousseau 2016 ; prior over-smoothing being the root of many problems ).

---

> ### Author Response · Authors · 2020-11-24
> **Clarifications**
>
> Thank you for your time in reviewing our work. We think that actually we are perhaps more well-aligned then you may think, and are sorry to see that this alignment was potentially obfuscated due to the paper’s presentation. It was not our intention to claim that we were introducing coverage to the machine learning community since, as you highlight, the topic has been a rich source of research for many decades. Instead, it was our intent to evaluate the empirical coverage properties of this specific class of methods since 1) they are the subject of much attention in the deep learning literature and 2) as the other reviewers note, such an evaluation for these models had not yet been done. We should have included the references provided by the reviewer and will provide them in updated versions of the manuscript.
>
> We would also like to push back against the claim that “these results will not generalize in any meaningful sense” since the results were consistent across all of the regression datasets on which the GP was used. We understand and appreciate that there are conditions (e.g. the ones from Hadji & Szabo) under which it can fail, but to discard 9 evaluations on real datasets under realistic conditions seems to be entirely too strong and unsupported by our results.

---

### Official Review · AnonReviewer4 · 2020-10-27
**Good experimental evaluation of uncertainty quantification (UQ) in deep learning, but questionable conclusion**

**Rating:** 4
**Confidence:** 4

**Review:**

Paper provides an  evaluation of the reliability of confidence levels of well known uncertainty quantification techniques in deep learning on classification and regression tasks. The question that the authors are trying to answer empirically is: when a model claims accuracy at a confidence level within a certain interval , how often does the actual accuracy   fall within  that interval?  This is conceptually similar to the recent slew of papers seeking to empirically evaluate the softmax calibration of deep models where the question there is how often do predicted probabilities of the winning class reflect the true probability of the correct answer, but in this paper the focus is on confidence level and confidence intervals.

Studies are conducted for both regression and classification.  Confidence levels and intervals are evaluated using the notion of coverage probability and width. While these have a straightforward interpretation in the regression setting, for classification the authors use the top K probabilities that captures 95% of the prediction probability mass to evaluate coverage and width. Thus for classification, the width is the number of classes over which 95% of the probability is smeared. Ideally one would want a model that has a low width, and high coverage probability (i.e a model that is both reliable and accurate). The aim of the paper is not to produce this ideal model, but rather to  empirically evaluate whether the predictive uncertainty of various methods proposed in the DL literature can be relied upon.  Various UQ methods are tested for both regression and classification datasets, and for the latter case, also under dataset shift.

Pros:
+ Paper is well written and ideas are, for the most part,  presented well.
+ Experiments test a variety of  state-of-the-art UQ methods.
+ There has not been work looking at this specific metric -- i.e., the reliability of prediction intervals. And with increasing usage of DL in high-risk applications, an evaluation of this kind might be useful.

Cons
- The authors appear to be conspicuously avoiding much usage of the terms "confidence levels" and "confidence intervals", but it appears that this is really what the paper is about.  Justify why you are taking this stance. The section on "theoretical coverage guarantees" is not sufficiently explanatory or convincing in this regard.

- A quantitative discussion on the mismatch between the coverage probability and the quality of softmax calibration is missing.

- My biggest concern is the conclusion of the paper: the authors state "we conclude that the methods we evaluated for
uncertainty quantification are likely insufficient for use in high-stakes, real-world applications where dataset shift is likely to occur." Yes, the models' coverage probabilities  are indeed significantly below the reported confidence level when data is corrupted (both for CIFAR-10 and ImageNet), but the fact that the width increases should give us an attack vector into the problem. You say this is not sufficient, but I'm not convinced this is the case.  95% of the probability mass is now smeared over a much larger number of classes. In other words, an increasing width necessarily means the predictions have increased in entropy, and also  that the probability mass in the winning class is now significantly lower under data corruption than what it was for the clean set. Both of these quantities (entropy and winning softmax) can be used to filter out predictions when the model is not confident (subject to a suitable confidence threshold), at least in the non-adversarial case. And in the real-world, this could be a practical approach to ascertain when a model's predictions should be trusted or discarded.

In summary, while the authors have done a commendable job with experimental evaluations, the conclusion is too strong and -- in my opinion -- incorrect to justify acceptance.

---

> ### Author Response · Authors · 2020-11-24
> **Confidence interval vs prediction interval**
>
> Thank you for your review -- we found it helpful and appreciate the time you and all other reviews contributed. With regards to specific terminology like “confidence levels” or “confidence intervals”, note that we are not actually attempting to construct a confidence interval as the term is used in the statistical literature. A confidence interval is an interval that, under repeated sampling, will contain the true population parameter (e.g. the mean) of interest with probability at least 1 - $\alpha$. Empirical evaluations of confidence intervals are only possible using simulations when the true value of the mean is known. Since such simulations would likely be unrealistically simple for deep learning scenarios, we sought here to evaluate prediction intervals which is a set that contains the observed values with probability 1 - $\alpha$. We appreciate this distinction is confusing and will make it more clear in a revised version of the paper.

---

### Official Review · AnonReviewer2 · 2020-10-29
**Interesting but not very insightful analysis, also lacks a methodological contribution.**

**Rating:** 4
**Confidence:** 4

**Review:**

**Summary and key claims**

This paper provides a comprehensive evaluation of the empirical frequentist coverage properties of existing uncertainty quantification baselines on both regression and classification tasks. The paper focuses on frequentist coverage as a faithful metric for the quality of uncertainty estimates. The experimental evaluations in the paper imply that accurate out-of-distribution coverage is a hard target for most existing baselines; a problem exacerbated by settings were dataset shifts are prevalent.

*The key contributions claimed by the paper are:*
- Introduces coverage and width as a natural and interpretable metrics for evaluating predictive uncertainty.
- Provides a comprehensive set of coverage evaluations for popular uncertainty quantification baselines.
- Examines how dataset shift affects these coverage properties.

**Originality and Significance**

Frequentist coverage is perhaps the most classical (and straightforward) measure of the quality of uncertainty estimates in statistics, so it's a bit odd that the authors claim the introduction of coverage and interval width as one of their key contributions. Despite not being as popular in the machine learning community, frequentist coverage has been considered in [R1] and [R2], and even coverage under dataset shifts was considered in [R3]. These existing papers not only consider frequentist coverage as a metric for uncertainty estimates, but they go as far as developing methods that provide theoretical guarantees on coverage. In fact, [R2] gives a more complete picture of uncertainty estimates by assessing both coverage and discriminative accuracy as both metrics do not necessarily correlate.

I think that the key contribution of the paper is the experimental evaluations on many baselines and many datasets to analyze the performance of different methods with respect to coverage. While this analysis is interesting, it lacked insights into baselines' performances and the role of the evaluation metric used in assessing the comparative performances of baselines. For the most part, the experimental section was limited to reporting performance of all baselines on all datasets without providing insights into **why some methods perform better than others w.r.t this specific coverage metric** and **how the introduction of the coverage metric changes our perception on which methods are best**. I was expecting to see more evaluations that rank baselines w.r.t say calibration or Brier score, and then show that a ranking based on coverage would be significantly different, thereby motivating the usage of coverage in the uncertainty analysis toolbox. I would have also appreciated a breakdown of aleatoric and epistemic uncertainty, and how coverage may be a good metric for assessing either types of uncertainties, etc. Having read the experimental section---which is the key section in this paper---I was not exactly sure what to make of it.

The key take away of the experiments highlighted in the abstract and discussion is that uncertainty estimates do not generalize well to out-of-distribution samples. However, such finding is not new and has been discussed before in (Ovadia et al. (2019)). Also, it is not clear how the introduction of the coverage metric helps us arrive at this conclusion; it seems to me that the same conclusion could have been arrived at with calibration or AUC-ROC on out-of-distribution samples.

**Technical comments**

I have two main comments on the technical aspects of the paper:

1) The authors found that GPs are clear winners when it comes to coverage. However, I am afraid that the frequentist coverage of Bayesian uncertainty (credible) intervals are extremely sensitive to the selection of the Bayesian prior (See the works of Szabo and van der Vaart in [R4] and references therein). Frequentist coverage is a specifically sensitive quantity in Bayesian analysis as a very large or very small prior length-scale of a GP kernel may give us very good or very bad coverage. The same issues are relevant (in a more subtle way) in Dropout NNs and any Bayesian NN approximation. Since most baselines considered in your frequentist analysis are actually Bayesian models, it is very important to report how robust are your findings to different selections of the priors (in this case, priors will correspond to hyperparameters). I did not find any discussion on the impact of hyperparameters in the resulting quality of uncertainty intervals and their impact on dataset shifts, despite this being a central concern in Bayesian models. A different approach for tuning hyperparameters may render models other than GPs come on top in your comparison.

2) Frequentist coverage is a concept associated with **regression** problems: we want a **contiguous** coverage set $C$ to contain the **real-valued** prediction target $y$ with a probability $1-\alpha$.

In $K$-classification problems, the true real-valued target is the class probabilities $p_K$, and a confidence set in this case would comprise a $K$-simplex that covers the true class probability $1-\alpha$ of the time. But the true class probabilities $p_k$ are never observed; we only observe discrete values for one out of K classes. So calculating empirical coverage of class probabilities is impossible in classification problems.

The authors extend the notion of coverage to classification in a different way: a coverage set $C$ is a discrete set of possible labels whose sum predicted probabilities add up to $1-\alpha$, and coverage is achieved if the true label belongs to this set (Equation (2)). I find this definition incomplete because your coverage set $C$ is not contiguous anymore; it may contain labels 1 and $K$ and excludes 2,...,$K-1$. As you can see, in this scenario a coverage set wouldn't make sense unless the targets 1 to $K$ are unordered. So I think you have to say that these applies only to unordered categorical targets for this to make sense.

Also, I do not see how this definition would work for binary classification, which is always ordered? In the case of binary classification, it seems to me that calibration is actually a more expressive metric than coverage as it accounts for class probability even when $K=2$.

**References**

[R1] Rina Foygel Barber, Emmanuel J Candes, Aaditya Ramdas, Ryan J Tibshirani, "Predictive inference with the jackknife+", arXiv, 2019.

[R2] Alaa, Ahmed M., and Mihaela van der Schaar. "Discriminative jackknife: Quantifying uncertainty in deep learning via higher-order influence functions." ICML (2020).

[R3]  Tibshirani, R. J., Barber, R. F., Candes, E., & Ramdas, A. (2019). Conformal prediction under covariate shift. In Advances in Neural Information Processing Systems (pp. 2530-2540).

[R4] Botond Szabó, A. W. van der Vaart, and J. H. van Zanten, Frequentist coverage of adaptive nonparametric Bayesian credible sets, Annals of statistics, 2015.

---

> ### Author Response · Authors · 2020-11-24
> **Literature and experiments**
>
> Thank you for your thorough and insightful review. We appreciate the references to the literature. While these references [R1, R2, R3] are related ideas, none of the work touch on the exact same concept we are trying to capture here. In [R1, R2], the authors develop techniques to create intervals that have provable coverage guarantees. While this is impressive work, here we aimed to quantify the empirical coverage of the built in uncertainty estimates of existing methods. To our knowledge, this has not been done before for the approximate Bayesian methods that we consider. However, we will include these citations in revisioned versions of the manuscript due to their relevance.
>
> With regards to the Experiments section, we will consider your suggestions for future iterations of the paper. Comparing the ordering of coverage against other metrics such as Brier score is a great idea. Indeed, we believe that coverage will strengthen the evidence that current methods do not behave as we might hope they would in OOD situations.
>
> Your comments about hyperparameters is particularly relevant and we will include a larger discussion about HPs in future iterations of the work. Our current study was designed to assess the performance of common setups that an end-user might employ in practice, and we do believe that it has captured that scenario. We did a hyperparameter search for the final models trained in the paper and with this many methods and datasets, an exhaustive analysis of the effect of all possible HPs on coverage would likely be imfeasible. But this remark is important and we will consider how to best incorporate it in a future iteration of our work.
>
> You are correct that there is an implicit assumption that the classification labels are unordered in our definition of coverage for classification. We will make that explicit in a future revision. However, for binary classification, is it true that labels are always ordered? In a dog vs. cat problem, for instance, there is not an inherent ordering.

---

### Official Review · AnonReviewer3 · 2020-11-01
**A timely survey of coverage properties, but visual communication of results needs more work.**

**Rating:** 4
**Confidence:** 3

**Review:**

## Summary
The authors compare empirical frequentist coverage of predictive intervals for several uncertainty quantification methods.  The paper covers both classification and regression.  The authors define an analogue of a confidence interval for classification.  Coverage properties are also studied under covariate shift between training and test sets.

## Pros
Coverage and width are a standard benchmarks for uncertainty quantification in statistics, and to my knowledge, this is the first work that undertakes a large-scale comparison for deep learning models.  Some inspiration seems to have been drawn from Ovadia et al. 2019 in that the set of methods compared are similar and the same architectures are used.  However, this work makes an important contribution in focusing on coverage / width, which I would agree are more interpretable metrics for practitioners.  The set of methods spans several important strains of the literature: ensembling, Bayesian approximation, Dropout, GPs.

## Cons
The work is timely and of broad interest; however, I think the presentation of results still needs some refining.  Given that this paper focuses on empirical results, I would suggest the authors spend more time developing effective visualizations to communicate their conclusions.

Tables 1 2 and 3 are perhaps necessary as a reference, but cry out for a visual aid.  The authors state "We see that higher coverage correlates with a higher average width."  This is something that seems like it could be communicated more immediately with the right plot.

Figure 4 conveys some visual trends, but also could be improved.  The coverage plot contains mostly blank space.  The dots are clustered together and impossible to differentiate.  In the width plot, what is communicated is that all methods have wider intervals with more shift.  However, it is hard to differentiate the methods: again, the dots are on top of each other, the colors blend such that they do not seem to actually correspond to the colors on the legend (perhaps alpha should not be used here?).

Figure 5 has similar problems.  The use of alpha means that colors blend together and cannot be looked up in the legend. As before, the dots in the legend are tiny, so it is hard to differentiate the shades even in the legend. What is visually communicated is the spread of performance, but conclusions about any particular method are nearly impossible.

Clearly this is a difficult problem to solve: there are 7 methods, and several axes of variation (coverage, width, shift).  However, the plots at the moment do not convey much information aside from overall trends. Visual understanding method-specific results is not possible at the moment.

I would also suggest that the authors devote more attention to the definition of coverage for predictive intervals, and how it relates to distribution shift.  For example, the authors define coverage as equation (1) holding for any distribution P.  It is not explicitly stated, but the implication here seems to be that the set $\hat{C}_n(x_n)$ is determined from training data distributed as P, and coverage is measured from data drawn from the same distribution (i.e. this definition does not allow covariate shift).  It would be useful for the authors to state in mathematical terms, what it means for coverage to hold under covariate shift.  Is this equivalent to the notion of conditional coverage as defined in Barber et al. 2020?  These are subtle enough concepts that I think they should be more precisely spelled out in the paper, even if some intuitive definition of covariate shift is widely understood. Clearly we cannot expect coverage to hold under a distribution shift that changes the conditional P(Y | X) between training and eval. What are the limits of what the authors allow?

I would also suggest that the authors might include an explicit analysis of conditional coverage.  For example, all methods seem to enjoy 95% coverage for in-distribution eval sets.  However, it would be interesting to know if this coverage is uniform across classes or any other useful clustering of the data.

### A few specifics:
*    The authors' analogue of confidence interval for a classifier is novel to me, and is a convenient way to unify the presentation of results between classification and regression.  If this is a novel definition, I would suggest the authors more explicitly point this out, as future literature may use it and should cite it.
*   Figure 4 is out of order with figure 3 - this needs to be fixed.

---

> ### Author Response · Authors · 2020-11-24
> **Figures are nontrivial**
>
> Thank you for your thorough and helpful review. We will rework the visual representation of results for a future version of the paper. Like you said, it is indeed a hard issue to convey these results with so many axes of variation.
>
> With regards to the specific definition of coverage under covariate shift, we will expand on this in future iterations. Right now, we consider “coverage under covariate shift” to be quantified in the same way as without shift: calculating the fraction of the test prediction interval/sets that contain the true value/label, which is not the same as conditional coverage in Barber et al. 2020. Indeed, nobody is expecting coverage properties to hold under heavy distribution shift. However, this work aims to understand empirically how rapidly coverage deteriorates under shift in practice, thus we did not specify what specific distribution shifts are permitted as in real life, that is unknowable.
>
> We will definitely consider an analysis of conditional coverage for future work. Thank you for the suggestion.

---

### Decision · Program_Chairs · 2021-01-07
**Final Decision**

**Decision:**

Reject

**Comment:**

Overall, the reviewers agree that there is definite value in the empirical evaluation you have provided. However, as you have acknowledged in your responses to the reviewers, the presentation could be significantly improved. A final point that was not touched upon by the reviewers--where possible (e.g. certainly not ImageNet, but for some of the smaller datasets in Table 1) it would be helpful to have a comparison to fully Bayesian methods (you have linear regression and GPs, but I don't see the implementation details; my suggestion is to implement these within an MCMC framework, specifying reasonable priors over the (hyper)parameters).